# YY2 Promotes Osteoblast Differentiation by Upregulating Osterix Transcriptional Activity

**DOI:** 10.3390/ijms23084303

**Published:** 2022-04-13

**Authors:** Meiyu Piao, Sung Ho Lee, Myeong Ji Kim, Hyung Sik Kim, Kwang Youl Lee

**Affiliations:** 1College of Pharmacy, Chonnam National University, Yongbong-ro, Buk-gu, Gwangju 61186, Korea; my101park@gmail.com (M.P.); puzim23@gmail.com (S.H.L.); audwl0815@gmail.com (M.J.K.); 2School of Pharmacy, Sungkyunkwan University, 2066 Seobu-ro, Jangan-gu, Suwon 16419, Korea

**Keywords:** YY2, Osterix, osteoblast differentiation

## Abstract

Yin Yang 2 (YY2) is a paralog of YY1, a well-known multifunctional transcription factor containing a C-terminal zinc finger domain. Although the role of YY1 in various biological processes, such as the cell cycle, cell differentiation and tissue development, is well established, the function of YY2 has not been fully determined. In this study, we investigated the functional role of YY2 during osteoblast differentiation. *YY2* overexpression and knockdown increased and decreased osteoblast differentiation, respectively, in BMP4-induced C2C12 cells. Mechanistically, *YY2* overexpression increased the mRNA and protein levels of Osterix (Osx), whereas *YY2* knockdown had the opposite effect. To investigate whether YY2 regulates *Osx* transcription, the effect of *YY2* overexpression and knockdown on *Osx* promoter activity was evaluated. *YY2* overexpression significantly increased *Osx* promoter activity in a dose-dependent manner, whereas *YY2* knockdown had the opposite effect. Furthermore, vectors containing deletion and point mutations were constructed to specify the regulation site. Both the Y1 and Y2 sites were responsible for YY2-mediated *Osx* promoter activation. These results indicate that YY2 is a positive regulator of osteoblast differentiation that functions by upregulating the promoter activity of *Osx*, a representative osteogenic transcription factor in C2C12 cells.

## 1. Introduction

Bone development is maintained by the formation and resorption of osteoblasts. Disruptions in the dynamic balance of formation and resorption, such as excessive resorption, can cause loss of bone mass, density, and strength, resulting in systemic diseases such as osteoporosis [1,2]. Therefore, promoting the formation of osteoblasts and inhibiting excessive resorption by osteoclasts are important, effective strategies for preventing and treating osteoporosis. However, the regulatory mechanism of bone formation and resorption remains under investigation.

Osteoblasts are the main cell type responsible for bone formation. They are derived from mesenchymal stem cells (MSCs), which differentiate into myoblasts, chondrocytes, adipocytes, or osteoblasts, depending on the conditions [3]. MSC differentiation into osteoblasts is controlled by several signaling pathways, including the bone morphogenetic protein (BMP), Wnt, and Notch signaling pathways, and by transcriptional regulators, mainly runt-related transcription factor (Runx2; Cbfa1/AML3), distal-less homeobox 5 (Dlx5), and Osterix (Osx) [4,5,6]. Particularly, Osx plays an important role in the proliferation and differentiation of osteoblasts. Osx, also known as Sp7, has three Cys2His2 (C2H2; Kruppel-type) zinc finger structures at the C-terminus and is highly homologous to the zinc finger structures of other speckled protein (SP) family members (such as Sp1 and Sp4) [7,8,9]. Osx-deficient mice exhibit failed bone formation and osteoblast differentiation and lack the expression of osteoblast-specific markers, such as bone sialoprotein (BSP) and osteonectin. In contrast, cells stably transfected with *Osx* clearly exhibit an osteoblastic phenotype, suggesting that Osx plays an important role in osteoblast differentiation. The mechanism of Osx function in osteoblast differentiation occurs via the binding of a GC-rich domain in the osteoblast marker gene promoter region [10]. *Osx* transcription is absent in *Runx2*-null mice. In contrast, the expression levels of *Runx2* in *Osx*-null mice are similar to those in wild-type mice, suggesting that *Runx2* is an important upstream regulator of *Osx* in the differentiation pathway [11].

Yin Yang 2 (YY2) is a member of the Yin Yang (YY) family and has a highly conserved DNA-binding zinc finger structural domain at the C-terminus. YY2 is highly homologous to another member of the YY family, Yin Yang 1 (YY1). YY2 may function as a transcriptional regulator, similar to YY1 [12,13]. However, unlike *YY1*, *YY2* is not ubiquitously expressed; therefore, the biological function of *YY2* has not been identified. Recently, YY2 has been shown to be involved in tumorigenesis as a novel tumor suppressor, with a role in cell cycle regulation and cell proliferation [14]. YY2 is also involved in the response to immune diseases, regulating the promoter activity of interleukin 4 (*IL-4*) by exerting an antagonistic effect that opposes the effect of YY1 [15]. During osteoblast differentiation, YY1 regulates *Runx2* transcriptional activity and inhibits osteoblast differentiation [16]. YY2 is reportedly involved in the pathogenesis of various diseases, but its role in osteoblast differentiation remains unclear.

Here, we present evidence of the role of YY2 in osteoblast differentiation. YY2 promoted BMP4-induced osteoblast differentiation by regulating the promoter activity of *Osx* and increasing *Osx* expression.

## 2. Results

### 2.1. YY2 Promoted BMP4-Induced Osteoblast Differentiation

To elucidate the role of YY2 in osteoblast differentiation, we first examined the effect of YY2 on BMP4-induced alkaline phosphatase (ALP) expression in C2C12 cells. ALP is an enzyme that is involved in bone mineralization so it is commonly used as an early marker for osteoblast differentiaion [17]. Under the influence of BMP4 in vitro, the myoblast cell line (C2C12) undergoes osteoblast differentiation [18]. Overexpression of *YY2* significantly enhanced BMP4-induced ALP activity in a dose-dependent manner (Figure 1A,B). We also examined the effect of YY2 on the expression of osteogenic markers. Overexpression of *YY2* increased the mRNA levels of ALP, osteocalcin (OC) after 2 days of treatment with BMP4 (Figure 1C,D). These results suggest that *YY2* overexpression promotes osteoblast differentiation.

### 2.2. Knockdown of YY2 Attenuated Osteoblast Differentiation

To confirm our findings, we examined the effect of *YY2* knockdown on osteoblast differentiation. The efficiency of the inhibition of endogenous YY2 was described previously [19]. First, *YY2* knockdown dramatically reduced BMP4-induced ALP activity (Figure 2A,B). Second, *YY2* knockdown significantly decreased the mRNA expression of osteoblast differentiation markers, such as ALP and OC after 2 days of treatment with BMP4 (Figure 2C,D). These results suggest that *YY2* knockdown inhibits osteoblast differentiation.

### 2.3. YY2 Postively Regulates the Promoter Activity of Osteoblast-Specific Marker Genes

To investigate whether *YY2* can increase the mRNA levels of osteogenic genes in a transcription level, we examined whether *YY2* could regulate the promoter activity of osteoblast-specific genes such as *ALP*, *BSP*, and *OC* using luciferase reporter, which contains each promoter region. Interestingly, YY2 enhanced the promoter activity of *ALP*, *BSP*, and *OC* in a dose-dependent manner (Figure 3A). In contrast, the knockdown of YY2 inhibited the promoter activity of *ALP*, *BSP*, and *OC* (Figure 3B). These results suggest that *YY2* positively regulates osteogenic genes in BMP4-induced osteoblast differentiation.

### 2.4. YY2 Upregulats the Expression of Osx

To examine the underlying mechanism of how YY2 does positively regulates the osteogenic gene transcription, we investigated the relationship between *YY2* and osteoblast differentiation-related transcription factors such as *Runx2* and *Osx*. First, the effect of YY2 on mRNA levels of *Runx2* and *Osx*. Of note, overexpression of YY2 increased the mRNA levels of *Osx* (Figure 4A). In contrast, the knockdown of *YY2* decreased the mRNA levels of *Osx* (Figure 4B). Interestingly, neither overexpression nor knockdown of *YY2* significantly affected the mRNA expression of *Runx2*. Overexpression of YY2 increased exogenous Osx but did not affect exogenous Runx2 (Figure 4C). Similarly, Overexpression of *YY2* enhanced endogenous Osx protein expression while it did not affect endogenous Runx2 protein expression during osteoblast differentiation (Figure 4D). These results indicate that *YY2* affects osteoblast differentiation by upregulating the expression of the specific transcription factor *Osx*.

### 2.5. YY2 Activates the Osx Promoter Activity

*Osx* exhibits specific expression during in vivo embryonic development and in vitro cell culture, indicating that *Osx* transcription is tightly controlled via promoter activity [20]. To investigate YY2-mediated regulation of *Osx* expression, we determined whether YY2 stimulates *Osx* transcription by using a 3.0 kb (from −3064 to −64) upstream regulatory sequence of the human *Osx* gene to analyze *Osx* promoter activity (Figure 5A). Overexpression of *YY2* significantly enhanced *Osx* promoter activity in a dose-dependent manner (Figure 5B). It was reported that YY2 and YY1 bind to the same DNA region, which contains the YY1 consensus motif (CGCCATnT) [21]. We found that the promoter region of *Osx* contained two YY1 consensus sequences, aacctCCATattccttgtc (Y1 site) and ctgctCCATttgctgagct (Y2 site) using genematix software. To determine whether the effect of YY2 on *Osx* occurs at the YY1 consensus motif, we used a reporter assay to analyze the effects of deletions in the 3.0 kb *Osx* promoter (Figure 5C). The promoter activity of the 1.4 kb (from −1441 to −64) and 0.7 kb (from −766 to −64), which contained a single consensus site, was significantly decreased compared to that of the 3.0 kb, which contained two consensus sites (Figure 5D). This suggests that the effect of YY2 on *Osx* occurs at the YY1 consensus site.

### 2.6. YY2 Regulates Osx Promoter by Binding YY1 Consensus Motif

To further demonstrate the importance of the YY1 consensus site for the YY2 function, we constructed each single-mutation plasmid (Y1-mt and Y2-mt) (Figure 6A). Overexpression of *YY2* exerted a significantly weaker effect on the mutant *Osx* promoters than on the wild-type promoters (Figure 6B). These results indicate that YY2-mediated upregulation of *Osx* expression stimulates *Osx* promoter activity at the YY1 consensus motif. The above results indicated that YY2 stimulates the *Osx* promoter at the YY1 consensus motif. To investigate whether YY2 could bind to the proposed binding site (Y2), we performed a ChIP assay using Myc antibody and a primer set flanking the binding site (Figure 6C). We found that YY2 binds Y2 sites on the *Osx* promoter (Figure 6D). These results suggest that YY2 acts as a novel regulator of *Osx* by binding to the YY1 consensus motif on the *Osx* promoter.

## 3. Discussion

Bone formation is divided into endochondral ossification and intramembranous ossification. These two different patterns of bone development involve a variety of transcription factors and secreted molecules, indicating the complexity of the bone formation process and the diversity of the regulatory processes [22]. Bone contains two specific cell types, osteoblasts, and osteoclasts. Osteoblasts are responsible for bone formation. Various cytokines, growth factors, hormones, signaling pathways, and specific transcription factors, such as Runx2 and Osx, are involved in the bone formation process by regulating the proliferation and differentiation of osteoblasts. Mice deficient in either Runx2 or Osx show a complete lack of intramembranous and endochondral ossification, demonstrating the importance of both transcription factors in osteoblast differentiation [11].

Sp7/Osx is an osteoblast-specific transcription factor that regulates bone formation and osteoblast differentiation in vivo and in vitro. Osx was concluded to be a downstream factor of Runx2 due to the absence of *Osx* expression in *Runx2*-null mice. Recently, BMP was found to induce *Osx* expression in a Runx2-dependent or non-Runx2-dependent manner [23,24]. *Osx* expression is also regulated by IGF [25], the inflammatory cytokine TNF-α [26], and parathyroid hormone [27]. This evidence suggests that the regulation of *Osx* expression by various signal transduction proteins and transcription factors is important for the regulation of osteoblast differentiation [28].

YY2 is a paralog of YY1 with a highly similar structure. Therefore, YY2 exhibits either transcriptional activation or transcriptional repression. Although the C-terminal region of *YY2* is similar to that of *YY1*, the N-terminal region is very different, suggesting that the YY2 protein functions differently from the YY1 protein [12,29,30]. YY2 recognizes the same DNA sequence as YY1 in vitro but exhibits different DNA binding affinities [21]. For example, interleukin (*IL*)*-4*, share the DNA binding sequence of YY1 and YY2 but have opposite biological functions [15]. To date, while the regulatory role of YY1 is well understood, the role of YY2 in cellular biological activity, particularly osteoblast differentiation, requires elucidation.

In the present study, our results emphasized the essential role of YY2 in osteoblast differentiation. *YY2* overexpression dramatically increased BMP4-induced osteoblast differentiation and stimulated mRNA expression of the osteoblast differentiation markers *ALP*, *OC,* and the transcription factors *Runx2* and *Osx*. YY2 also significantly upregulated the promoter activity of *ALP*, *BSP*, and *OC*. Therefore, YY2 and YY1 exhibited opposite phenotypes during osteoblast differentiation. To clarify the molecular mechanisms of YY2-mediated promotion of osteoblast differentiation, we further investigated the relationship between YY2 and osteoblast transcription factors. YY2 specifically stimulated the expression of the transcription factor *Osx* during osteoblast differentiation. While YY2 did not affect both mRNA and protein levels of *Runx2*.

Based on our hypothesis that YY2-mediated gene regulation is promoter-dependent, we constructed a luciferase reporter plasmid containing the human *Osx* promoter. In agreement with our hypothesis, YY2 upregulated *Osx* promoter activity in a dose-dependent manner. Using Genomatix software, we identified two YY1 consensus sequences located in the *Osx* promoter region. Two YY1 consensus sequences were identified as essential for YY2-mediated regulation of the *Osx* promoter. More importantly, YY2 regulated *Osx* promoter activity in a YY1-independent manner. Subsequent deletions in the *Osx* promoter revealed that the reduction in promoter activity at the Y1 site in response to *YY2* overexpression was significantly smaller than that in the wild-type promoter. Nevertheless, the presence of the Y2 site allowed the activity of the deleted promoter to be maintained. This suggests that the two YY1 consensus sequences are essential for YY2-mediated regulation of the *Osx* promoter.

To further demonstrate the importance of the consensus sequences, mutations were introduced into the Y1 and Y2 sites, separately and simultaneously. Consistent with our hypothesis, the promoter activity associated with mutations in the Y1 or Y2 sites was significantly reduced. Furthermore, we found that YY2 binds to the Y2 site in the promoter region of Osx, but not the Y1 site. The mutated Y1 site loses the induction of YY2 and we retain the possibility that there are other factors on the Y1 site involved in the regulation of the Osx promoter by YY2 that require further elucidation. These findings highlighted the importance of the YY1 consensus motif for YY2-mediated regulation of the *Osx* promoter, and we determined that YY2 regulated *Osx* promoter activity and expression by binding to the YY1 consensus motif, especially the Y2 site. Moreover, *YY2* knockdown decreased ALP activity and osteogenic gene expression, resulting in the suppression of osteoblast differentiation.

In summary, we provide novel evidence of the essential role of YY2 in osteoblast differentiation. Our findings also indicate that *Osx* is a major target of YY2. YY2 increased osteoblast differentiation by binding with the *Osx* promoter to regulate promoter activity and gene expression.

## 4. Experimental Procedures

### 4.1. Cell Cultures and Transfection

Mouse pre-myogenic C2C12 and human embryonic kidney 293 (HEK293) cells were cultured in 5% CO_2_ at 37 °C and maintained in Dulbecco’s modified Eagle medium (DMEM; Gibco, Carlsbad, CA, USA) containing 10% FBS (Welgene Inc., Daegu, Korea) and 1% antibiotic-antimycotic solution (Gibco). Transient transfections were performed using the polyethyleneimine (PEI)-mediated method (Polysciences, Warrington, PA, USA), and transfection controls were established using empty vectors.

### 4.2. Alkaline Phosphatase (ALP) Staining

C2C12 cells were cultured for 72 h in 2% FBS medium containing BMP4 to induce differentiation. The cells were washed twice with phosphate-buffered saline (PBS), fixed in 4% paraformaldehyde at room temperature (RT) for 15 min, washed twice again with PBS, and then incubated in BCIP/NBT substrate (Sigma-Aldrich, St. Louis, MO, USA) for 15 min at RT. Quantification of ALP staining was performed at an absorbance of 480 nm.

### 4.3. Luciferase Assay

C2C12 cells were seeded in 24-well plates at a density of 2 × 10^4^ cells/well. The cells were transfected with a CMV promoter-driven β-galactosidase reporter gene (pCMV-β-gal) (0.05 μg) and luciferase reporter gene (0.3 μg), and the cell lysates were analyzed for luciferase activity 36 h after differentiation. A luciferase reporter gene assay kit (Promega, Madison, WI, USA) was used to measure luciferase activity on the TriStar² Multimode Reader platform. All experiments were performed in triplicate.

### 4.4. RNA Isolation and RT-qPCR

Total mRNA was extracted from cultured C2C12 cells using RNAiso Plus (Total RNA Extraction Reagent; TaKaRa, Tokyo, Japan). cDNA was synthesized from 1 μg of total RNA using Oligo dT primers and reverse transcriptase (Promega, Madison, WI, USA). cDNA was synthesized using SYBR Premix Ex Taq kit (TaKaRa) on a CFX96 real-time PCR system(Bio-Rad Laboratories, Hercules, CA, USA). The following PCR conditions were used: initial denaturation at 94 °C for 3 min, 40 cycles of denaturation at 94 °C for 30 s, and annealing at the optimal temperature for each primer pair for 30 s. The primer sequences used for PCR were as follows: mALP forward 5′-ATC TTT GGT CTG GCT CCC ATG-3′ and reverse 5′-TTT CCC GTT CAC CGT CCA C-3′; mOC forward 5′-GCA ATA AGG TAG TGA ACA GAC TCC-3′ and reverse 5′-GTT TGT AGG CGG TCT TCA AGC-3′; mRunx2 forward 5′-CCT GAA CTC TGC ACC AAG TCC T-3′ and reverse 5′-TCA TCT GGC TCA GAT AGG AGG G-3′; mOsx forward 5′-TCG CAT CTG AAA GCC CAC TT-3′ and reverse 5′-CTC AAG TGG TCG CTT CTG GT-3′; GAPDH forward 5′-AGG TCG GTG TGA ACG GAT TTG-3′ and reverse 5′-GGG GTC GTT GAT GGC AAC A-3′.

### 4.5. Immunoblotting

Cells were washed twice with PBS and lysed in ice-cold lysis buffer (25 mM HEPES [pH 7.4], 150 mM NaCl, 1% NP-40, 0.25% sodium deoxycholate, 10% glycerol, 25 mM sodium fluoride, 1 mM EDTA, 1 mM Na_3_VO_4_, 250 μM PMSF, 10 μg/mL leupeptin, 10 μg/mL peptidase, and 10 μg/mL aprotinin) for 15 min, followed by centrifugation at 13,000 rpm and 4 °C. The supernatant was then removed. Equal amounts of protein were subjected to sodium dodecyl sulfate-polyacrylamide gel electrophoresis and transferred to polyvinylidene difluoride membranes (Immobilon-P, Millipore, Burlington, MA, USA). The membranes were blocked with 5% skim milk and incubated with the appropriate antibodies against Runx2 (sc-390351, Santa Cruz Biotechnology, Santa Cruz, CA, USA), Osx (sc-393325, Santa Cruz Biotechnology), Myc (9E10, Roche Applied Science, Branchburg, NJ, USA) (Santa Cruz Biotechnology), HA (12CA5, Roche Applied Science), and α-tubulin (sc-8035, Santa Cruz Biotechnology). After washing and incubation with the appropriate horseradish peroxidase (HRP)-conjugated mouse and rabbit secondary antibodies, visualization was performed using Immobilon Western Chemiluminescent HRP Substrate (WBKLS0500, Millipore) on the Amersham^TM^ ImageQuant^TM^ 800 biomolecular imager (GE Healthcare Life Sciences, Marlborough, MA, USA).

### 4.6. Construction of Human Osx Promoters with Deletions and Point Mutations

A 3.0 kb 5′ flanking region was obtained by PCR, using the following primer: 5′-GGG GTA CCC TCC GAG ATA GTA GGG TTC G-3′. Serial 5′ truncation of this region was performed by PCR with a sense primer (for the 1.4 kb promoter, 5′-GGG GTA CCC CAA AGA GGT TAG GTG TCG GC-3′; for the 0.7 kb promoter, 5′-GGG GTA CCC CTT TTC CCT CCT GTT CCT TC-3′) and a standard antisense primer, 5′-CCG CTC GAG GGG GGG TAG AGA GAG AAA AGG-3′. To mutate the putative YY1 consensus sites in the 3.0 kb promoter, mutant primers for aac ctC CAT att cct tgt c (sense: 5′-GGT AAC TGA CAA GGA ATC TTG AGG TTA GG-3′; antisense: 5′-CCT AAC CTC AAG ATT CCT TGT CAG TTA CC-3′) and ctg ctC CAT ttg ctg agct (sense: 5′-GAG CTC AGC AAC TTG AGC AGG AAA TTT GG-3′; antisense: 5′-CCA AAT TTC CTG CTC AAG TTG CTG AGC TC-3′) were applied in combination with the primers used to amplify the normal 3.0 kb promoter. The overlap-extension PCR method was used for mutagenesis. Amplicons were cut with KpnI and XhoI restriction endonucleases and incorporated into a pGL3-basic vector cut with the same enzymes, resulting in several promoter-firefly luciferase constructs.

### 4.7. Chromatin Immunoprecipitation (ChIP)

ChIP tests were carried out according to the manufacturer’s instructions using a commercial kit (#117-295, Millipore, Burlington, MA, USA). Chromatin samples were prepared from HEK 293 cells at the indicated time after transfection of Myc-empty or Myc-YY2. Briefly, HEK 293 cells were cross-linked with formaldehyde and incubated for 15 min at 37 °C, and the cell lysates were treated with SDS lysis buffer prior to sonication to shear DNA into 200–1000 bp fragments. Protein cross-linked with DNA was then immunoprecipitated with Myc antibody or anti-IgG antibody (sc-2027; Santa Cruz Biotechnology) and add 60 μL of protein A Agarose/Salmon Sperm DNA. After wash and elution, chromatin was then reverse-crosslinked for 4 hr at 65 °C, and extracted DNA was used as a template for PCR. The following ChIP PCR primers were used: YY2 binding site in the Osx promoter (Y2-site), forward 5′-CTC CCT GAT CCC TTC TTT GG-3′ and reverse 5′-AAG GCT CCA GAT CCA ATG AG-3′

### 4.8. Knockdown Experiment

Small interfering RNA oligonucleotides were generated by targeting a 22 base sequence, GTA GTA GAG ATC ATG ATA A, of the mouse *YY2* gene. Sense and antisense oligonucleotides were annealed and ligated into a pSuper-retro vector (Oligoengine, Seattle, WA, USA).

### 4.9. Statistical Analysis

All experiments were repeated at least twice with three independent, generating qualitatively identical results. The results are given as the mean ± standard error of the mean. The student’s *t*-test was used to analyze the data, with *p* < 0.05 indicating significance.

## Figures and Tables

**Figure 1 ijms-23-04303-f001:**
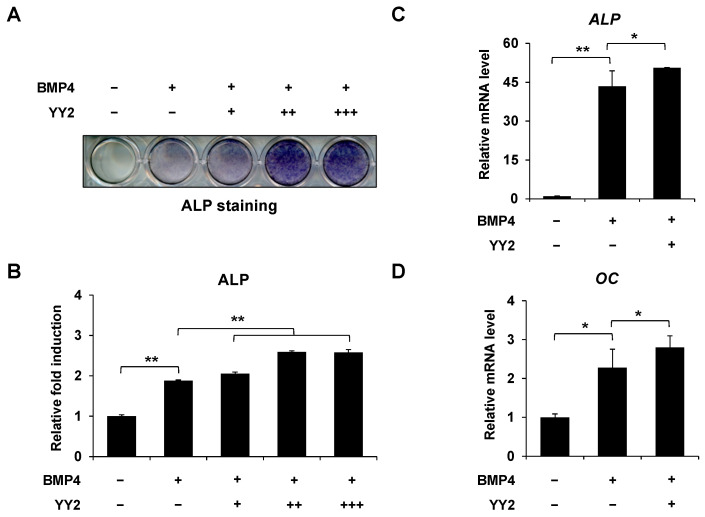
YY2 promotes BMP4-induced osteoblast differentiation. (**A**,**B**) Overexpression of YY2 significantly enhanced BMP4-induced expression of ALP in a dose-dependent manner. ALP staining and quantitative analysis of ALP activity after empty vector or increasing amounts of YY2 transfected C2C12 cells treated with or without BMP4 for 72 h. (**C**,**D**) Overexpression of YY2 increased the mRNA levels of osteogenic marker genes such as ALP and OC.YY2 or empty vector-transfected C2C12 cells were stimulated with BMP4 for 48 h. The expression of ALP, OC is compared by RT-qPCR. Data are expressed as mean ± SEM of at least three experiments. * *p* < 0.05, ** *p* < 0.01.

**Figure 2 ijms-23-04303-f002:**
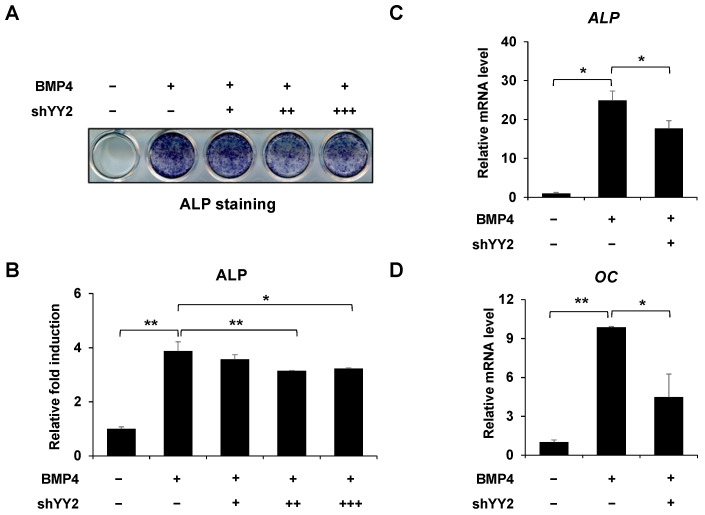
Knockdown of YY2 attenuates osteoblast differentiation. (**A**,**B**) Knockdown of YY2 reduced osteoblast differentiation. ALP staining and quantitative analysis of ALP activity after shControl (pSuper-empty vector) or increasing amounts of shYY2 transfected C2C12 cells treated with or without BMP4 for 72 h. (**C**,**D**) Knockdown of YY2 decreased the expression of osteoblast markers such as ALP and OC. C2C12 cells were transfected with shControl (pSuper-empty vector) or shYY2 and then stimulated with BMP4 for 48 h. The expression of ALP and OC is analyzed by RT-qPCR. Data are expressed as mean ± SEM of at least three experiments. * *p* < 0.05, and ** *p* < 0.01.

**Figure 3 ijms-23-04303-f003:**
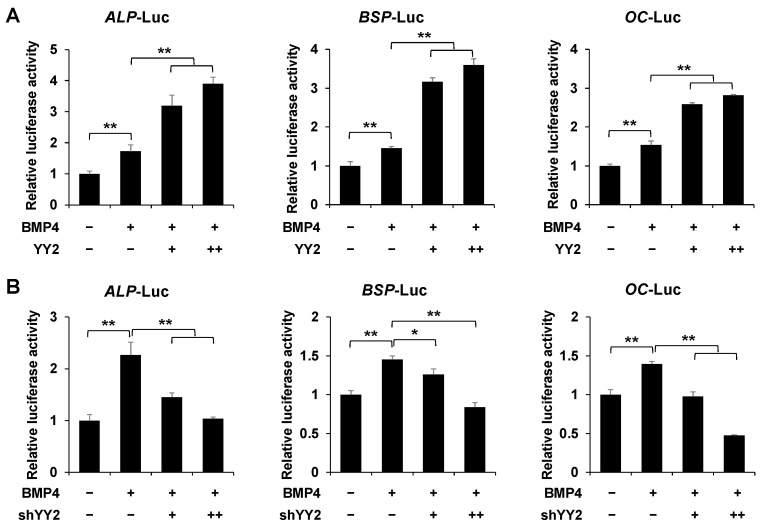
YY2 induces the expression of osteoblast-specific marker genes. C2C12 cells were transfected with ALP, BSP, and OC-Luc reporter genes (0.3 μg), and β-gal (0.05 μg), and increasing amounts of YY2 or shYY2, then treated with or without BMP4 for 36 h. (**A**) YY2 enhanced the promoter activity of ALP, BSP, and OC. (**B**) Knockdown of YY2 decreases the expression of osteoblast-specific reporters. Data are expressed as mean ± SEM of at least three experiments. * *p* < 0.05 and ** *p* < 0.01.

**Figure 4 ijms-23-04303-f004:**
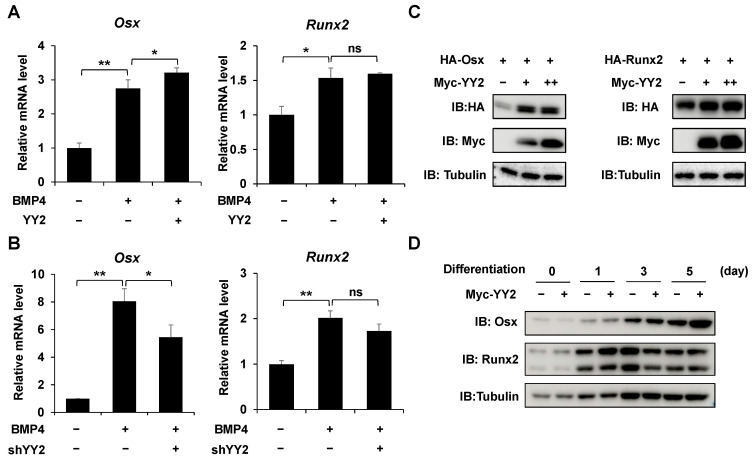
YY2 upregulates the expression of Osx. (**A**) Overexpression of *YY2* increased the mRNA levels of *Osx*. *YY2* or empty vector-transfected C2C12 cells were stimulated with or without BMP4 for 48 h. (**B**) Knockdown of YY2 decreased the mRNA levels of *Osx*. Knockdown of *YY2* or shCon (pSuper vector) transfected C2C12 cells were stimulated with or without BMP4 for 48 h. The expression of Runx2 and Osx is compared by RT-qPCR. Data are expressed as mean ± SEM of at least three experiments. * *p* < 0.05, ** *p* < 0.01, and ns: non-significant. (**C**) Overexpression of *YY2* increases the protein level of Osx. HEK293 cells were transfected with HA-Osx (left panel) or HA-Runx2 (right panel) and increasing amounts of Myc-YY2. After 48 h, the cell lysates were measured by Western blot. Tubulin was used as the loading control. (**D**) Overexpression of YY2 enhanced the protein levels of Osx during osteoblast differentiation. The numbers indicate the time points of differentiation induction. The targeted proteins were detected by western blotting.

**Figure 5 ijms-23-04303-f005:**
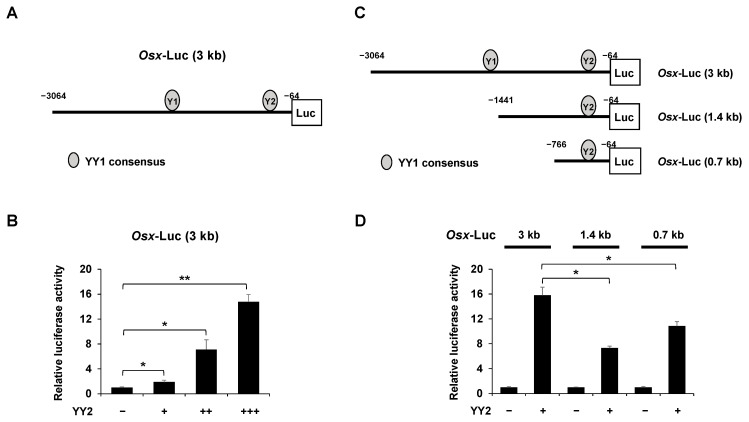
YY2 activates the *Osx* promoter activity. (**A**) Schematic diagram of pGL3 vector constructed human *Osx* promoter (from −3064 to −64). HEK 293cells were transfected with the indicated reporter gene {*Osx*-Luc (3 kb), *Osx*-Luc (1.4 kb), or *Osx*-Luc (0.7 kb)} (0.3 μg) and β-gal (0.05 μg) and YY2 expression plasmid. After 48 h, luciferase activities were measured. (**B**) The effect of YY2 on *Osx* promoter activity. Overexpression of YY2 significantly enhanced Osx promoter activity in a dose-dependent manner. (**C**) Schematic diagram of pGL3 vector constructed with the indicated range of *Osx* promoter region (3 kb, 1.4 kb, and 0.7 kb size). (**D**) The effect of YY2 on *Osx* promoter activity in the different *Osx* promoter (3 kb, 1.4 kb, and 0.7 kb size). YY2-activated *Osx* promoter activity was significantly attenuated in *Osx*-Luc (1.4 kb and 0.7 kb) compared to *Osx*-Luc (3.0 kb). Data are expressed as mean ± SEM of at least three experiments. * *p* < 0.05 and ** *p* < 0.01.

**Figure 6 ijms-23-04303-f006:**
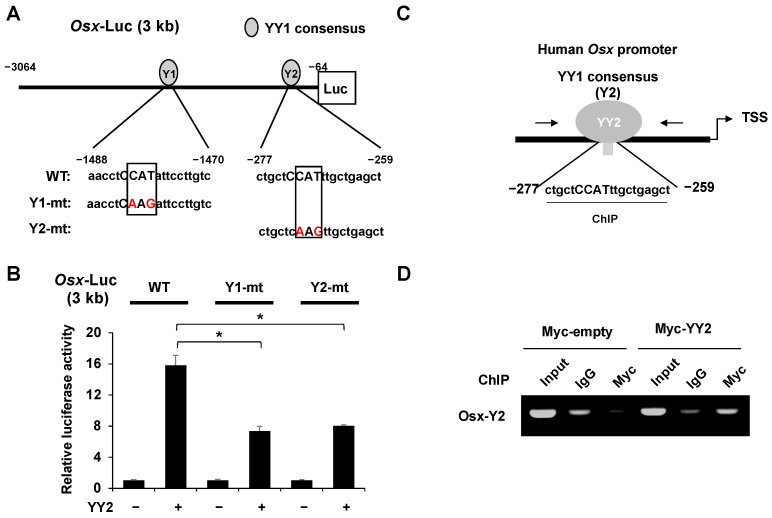
YY2 regulates the *Osx* promoter by binding to the YY1 consensus motif. (**A**) Schematic diagram of pGL3-*Osx* promoter mutant forms at YY1 binding site. (**B**) The effect of YY2 on the indicated *Osx* promoter activity. HEK 293cells were transfected with Osx-Luc (WT, Y1-mt, and Y2-mt) (0.3 μg) and β-gal (0.05 μg) and YY2. After 48 h, luciferase activities were measured. YY2-activated Osx promoter activity was significantly attenuated by mutations (Y1-mt and Y2-mt). Data are expressed as mean ± SEM of at least three experiments. * *p* < 0.05. (**C**) Schematic diagram of ChIP for Y2 site of *Osx* promoter and YY2. (**D**) ChIP assay was performed on HEK 293 cells using IgG and Myc antibodies. The immunoprecipitated DNA was used as a PCR template to detect the Y2 site of Osx promoter region.

## Data Availability

Please contact the corresponding author for reasonable data requests.

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
