# Peer review of "YY2 Promotes Osteoblast Differentiation by Upregulating Osterix Transcriptional Activity"

_ijms, 2022, doi:10.3390/ijms23084303_

Round 1
Reviewer 1 Report
In this paper it has been demonstrated that YY2 overexpression increases osteoblast differentiation and protein levels of osterix while YY2 knockdown has the opposite effect.
The paper is well-written, conducted experiments are described in detail while results are clearly presented.
Author Response
We really appreciate your comments.
Reviewer 2 Report
Dear Authors,
The article titled “YY2 Promotes Osteoblast Differentiation by Upregulating Osterix Transcriptional Activity”, submitted to the International Journal of Molecular Sciences is in the scope of the journal. In my opinion, the manuscript presents unique results about the role and the mechanism of activity of YY2 transcription factor in osteoblast differentiation. The research methods and statistical analysis are chosen correctly, the descriptions of used methods are detail to be reproducible. Although the presentation of results is good quality, I find that it should be indicated, e.g. in caption of Y axis, that relative fold induction is concerned with ALP (Fig.1 and 2). Moreover, in my opinion the statements in lines 244 and 245 (Discussion) are a little bit contradictory with results descriptions presented in lines 137 -143.
Author Response
We really appreciate your comments. Please see the attachment regarding "Response to your comments".
